# Deepfakes: we need to re-think the concept of "real" images.

## Abstract

The wide availability and low usability barrier of modern image generation models has triggered the reasonable fear of criminal misconduct and negative social implications. The machine learning community has been engaging this problem with an extensive series of publications proposing algorithmic solutions for the detection of "fake", e.g. entirely generated or partially manipulated images. While there is undoubtedly some progress towards technical solutions of the problem, we argue that current and prior work is focusing too much on generative algorithms and "fake" data-samples, neglecting a clear definition and data collection of "real" images.

The fundamental question *"what is a real image?"* might appear to be quite philosophical, but our analysis shows that the development and evaluation of basically all current "fake"-detection methods is relying on only a few, quite old low-resolution datasets of "real" images like *ImageNet*. However, the technology for the acquisition of "real" images, aka taking photos, has drastically evolved over the last decade: Today, over 90% of all photographs are produced by smartphones which typically use algorithms to compute an image from multiple inputs (over time) from multiple sensors. Based on the fact that these image formation algorithms are typically neural network architectures which are closely related to "fake"-image generators, we state the position that today, **we need to re-think the concept of "real" images**.

The purpose of this position paper is to raise the awareness of the current shortcomings in this active field of research and to trigger an open discussion wether the detection of "fake" images is a sound objective at all. At the very least, we need a clear technical definition of "real" images and new benchmark datasets.

*"Reality is captured in the categorical nets of language only at the expense of fatal distortion."*

FRIEDRICH NIETZSCHE

## Introduction

The fundamental question *"- what is real?- "* has been explored by generations of philosophers and has led to key philosophical perspectives like *realism*, *idealism* and *skepticism*, involving prominent contributors such as *Hegel*, *Kant* or *Nietzsche*. In context of modern generative models, the slight alteration *"- is it real? -"*, when confronted with images or voice transmissions has led to increasing concerns regarding the social and criminal impact of so-called Deepfakes [27, 26]. So far, the machine learning community has mostly by-passed the complex philosophical perspectives on reality and operated with a very simple technical definition in order to design Deepfake detectors: data is "real" when its not "fake", and data is "fake" when it has been generated or altered by

(generative) algorithms instead of being "recorded" from the "real word"... While many philosophers probably would have strongly disagreed with this simplification all along, it appeared to be quite sufficient for technical approaches towards Deepfake detection. **However, we argue that this is not true (anymore)**.

We focus our main argument on the detection of "fake" images and point out that the above ad-hoc definition also fails in a purely technical sense: over $90\%$ [45] of all photographs that are "taken" today are produced with smart phone cameras. In contrast to traditional cameras, modern phones do not solely capture images as 2D lens projections of the real 3D world. Instead, they typically apply complex image enhancement algorithms and fuse multiple (camera) sensor outputs from different cameras and capture times [8, 29]. Hence, modern image devices are actually computing images, rather than "taking" them. This raises two important questions in the context of Deepfake detection:

◇ The first one is purely technical and simply asks: *"do our current fake detection datasets and benchmarks appropriately account for this technical development of imaging?"* - we will show that the clear answer to this is **NO**.

◇ The second, the more philosophical question is: *"if all images are processed by algorithms, how do we actually define real images?"* - or in even more skeptic (in the philosophical sens) terms: *"can we even clearly define what a real image is?"*

In summary, we derive the following position:

> ## Position: If we want to find a technical solution towards *DeepFake* detection, *we need to re-think the concept of "real" images.*
>
> To support our broader position, we follow a sequence of propositions supported by empirical and theoretical evidence to form our line of argumentation:
>
> - **proposition #1: Current "fake" detection benchmarks and datasets provide insufficient "real" samples.**
>
> - **proposition #2: Even the latest publicly available collections of "real" images contain mostly outdated and pre-processed low-resolution images.**
>
> - **proposition #3: Modern imaging devices like mobile phones are computing rather than "taking" photos. "Real" training sets must contain such samples.**
>
> - **proposition #4: The omnipresence of (automatic) image enhancing algorithms in modern imaging devices requires a redefinition of "real" images.**
>
> For the sake of clarity and simplicity, we focus our argumentation on the sub-field of **generated image detection** [25], but our positions can be transferred to other sub-fields of Deepfake image detection without loss of generalization.

**Detecting generated images - a brief review.**

While the actual Deepfake detection algorithms are not the main subject of our position, for the sake of completeness we start with a brief, hence incomplete, overview of current detection approaches. Works on the detection of generated images have begun wright after the introduction of Generative Adversarial Networks (GAM) [15]. Early attempts tried to locate image inconsistencies like wrong shadows or reflections [31], but most approaches have been utilizing established **image feature extractors**. For example, Wang et.al [42] use a simple supervised classification approach with a standard ResNet-50 [17] CNN as feature extractor, which provides surprisingly good detection performance for known generators (but rather poor generalization to unseen generators). Other approaches based on spatial features include the use of co-occurrence matrices [30], classification on pre-trained VLM feature spaces [32, 40] and Student-Teacher setups [50].

A complementary approach is to explore the **frequency space representation** of images: [11] showed that the band-limited up-convolution in CNN based GANs induces easy detectable frequency artifacts. In a similar approach, [13] used a Discrete Cosine Transformation (DCT) instead of Discrete Fourier Transformation (DFT) features. Further approaches include the detection of diffusion reconstruction errors [43], patch based analysis [3, 2, 49] and online methods [12].

Despite the very high detection rates which are typically reported for detection algorithms, many approaches are known to suffer from poor generalization to unknown models [37, 32, 50] and insufficient robustness against image augmentation/pre-processing [47]. Grommelt et. al [16] showed strong biases in existing benchmarks toward JPEG compression (real: compressed, fake uncompressed) and image shapes (real arbitrary sizes, fake: fixed square).

**Propositions**

> **Proposition #1**
>
> **Current "fake" detection benchmarks and datasets provide insufficient "real" samples:**
> - Dataset mostly contain older images which were taken way before modern image enhancement algorithms have become standard practice on imaging devices.
> - Datasets mostly contain only low-resolution images.
> - Datasets are widely biased in terms od image shapes and compression.

Table 1 gives an overview of **current generated image detection datasets** used in benchmarks evaluating state-of-the-art (SOTA) "fake-detection" algorithms. While the creators of all listed datasets spend a lot of effort to provide new and diverse "fake" image samples, typically generated by large numbers of different generation algorithms, they are surprisingly relying on a very limited number of the very same, rather old datasets to cover the "real" samples.

| "Real" data source | GenImage [50] | Wang [42] | DIRE [43] | Fake or JPEG [16] | Epstein et al. [12] | Ricker et. al. [37] | Ojha et al. [32] | CIFAKE [1] | UniversalFakeDetect [32] | WildFake [18] | 2022 IEEE Cup [6] | De-Fake [40] | DiffusionDB** [44] | ArtiFact [35] | Chameleon [47] | count |
|---|---|---|---|---|---|---|---|---|---|---|---|---|---|---|---|---|
| Cifar 10 [22] | | | | | | | | ✓ | | | | | | | | 1 |
| ImageNet [9] | ✓ | ✓ | ✓ | ✓ | | | ✓ | | | ✓ | ✓ | | | ✓ | | **8** |
| MS-Coco [23] | | ✓ | | | | | | | | ✓ | ✓ | ✓ | | | | 3 |
| Flickr30k [34] | | | | | | | | | | | | ✓ | | | | 1 |
| CelebA [24] | | ✓ | | | | | | | | | | | | | | 1 |
| LSUN [48] | | ✓ | ✓ | ✓ | | ✓ | | | | ✓ | ✓ | | | ✓ | | 7 |
| CelebA-HQ [20] | | | ✓ | ✓ | | | | | | ✓ | | | | ✓ | | 3 |
| FFHQ [21] | | | | | | | | | | ✓ | ✓ | | | ✓ | | 3 |
| AFHQ [4] | | | | | | | | | | ✓ | | | | ✓ | | 2 |
| LHQ [41] | | | | | | | | | | | | | | ✓ | | 1 |
| LAION 400-M [39] | | | | | ✓ | | ✓ | | ✓ | | | | | | | 3 |
| Re-LAION 5B [38] | | | | | | | | | | | | | | | | 0 |
| unsplash.com | | | | | | | | | | | | | | | ✓* | 1 |

Table 1: Generated image detection datasets and benchmarks. See table 2 for details on the data sources. Notes: *20k random samples, **dataset does not contain "real" images.

A closer investigation of these "real" datasets in table 2 reveals that most of these have been **created and published ten to fifteen years ago**. Even for the latest collections from 2022 one has to note that the majority of images actually has been taken way before the date of the dataset publication (see proposition #2). We argue that this reliance on older data samples is very likely to cause significant problems towards the generalization of these benchmark results to fake detection

applications "in the wild":

I) The default usage of image enhancement and fusion algorithms in modern imaging devices is not represented in this data, which likely leads to weak generalization (aka false positives). We show evidence for this concern in the discussion of proposition #3.

II) Low image resolution with limited size ratios can also induce unintended biases into the detector training. [16] showed that size-biases are exploited by detectors, leading to weak generalization.

III) All investigated datasets exclusively contain JPEG compressed images, which also has been shown [16] to heavily bias detectors.

| Dataset Name | Year of Creation | # Images |
|---|---|---|
| Cifar 10 [22] | 2009 | 60,000 |
| ImageNet [9] | 2009 | 1,281,167 |
| MS-Coco [23] | 2014 | 328,000 |
| Flickr30k [34] | 2015 | 30,000 |
| RAISE [7] | 2015 | 8,156 |
| CelebA [24] | 2015 | 202,599 |
| LSUN [48] | 2015 | 120,000,000 |
| CelebA-HQ [20] | 2018 | 30,000 |
| FFHQ [21] | 2019 | 70,000 |
| AFHQ [4] | 2020 | 15,000 |
| LHQ (landscape) [41] | 2021 (2015*) | 90,000 |
| LAION 400-M [39] | 2021 | 400,000,000 |
| Re-LAION 5B [38] | 8/2024 (2022**) | 5,526,641,167 |

Table 2: Data sources for "real" images used by the benchmarks listed in table 1 by year of creation and number of contained images. Notes: *LHQ is created from a subset of Flickr30k. **Re-LAION 5B is a sanitized re-publication of the 2022 LAION 5B dataset.

> **Proposition #2**
>
> **Even the latest publicly available collections of "real" images contain mostly outdated and pre-processed low-resolution images.**

A logical consequence of our argumentation in proposition #1 would be to rely on "real" images from the latest available datasets for detector training and evaluation. Unfortunately, our following analysis of the most recent and most comprehensive public **LAION-5B** [38] image dataset shows that there is currently no sufficient "real" data available at scale. We focus our analysis of Re-LAION 5B on images taken by mobile phones, more specifically *Apple iPhones*. This is motivated by the fact that over 90% [45] of all images have been taken with mobile phones in recent years. We consider *iPhones* to be a good proxy for modern imaging devices in general because of their dominant market share[2], technology leadership and easy identifiability in the LAION metadata. Figure 1 shows the phone model and image creation statistics of the full LAION-5B data-set, while figure 2 displays the 2d histogram of image resolutions. Given this analysis, we draw the following concussions:

IV) The mean age of images in this very latest dataset is about 7 to 8 years, with only a few samples taken after 2020 and even fewer by modern *iPhone* models. This excludes most modern image enhancement and fusion algorithms from the dataset.

V) While the diversity of image shapes is much higher than in previous datasets, LAION 5B still provides mostly low-resolution images with strong biases to certain aspect rations. Given

---

[1]The low counts for *iPhones-9* and *iPhones-10* models are most likely due to a changes in *exif*-tag IDs by iOS for these models and does not generally affect the core findings of our analysis.

[2]around 20% over the last 10 years [36]

the imaging resolutions available in the *iPhones* at the image creation time, we can only conclude that most of the image have been down-sampled or otherwise post-processed.

**Taking a glance at current image enhancement and fusion algorithms.**

Since the main argument of our next proposition is based on the recent advancement of image enhancement and fusion algorithms, which are now commonly applied on mobile imaging devices, we shell give a brief overview here. Please note that due to size limitations and based on the goal not to interrupt the flow of arguments, we neither aim for completeness nor technical depth in the following interlude. The interested reader may be referred to [8] or [29] for comprehensive reviews of image enhancement methods on mobile devices.

In a nutshell, the main objective of mobile image enhancement is to try to algorithmically compensate the strong physical limitations of the rather small cameras phones. Due to the strong size constraints, mobile cameras have low apertures and small imaging planes (chip sizes) which both strongly limit their light efficiency. The accompanying low focal ranges also contradict the wide range of intended use cases, which typically last from taking panorama pictures to near field QR-code scanning [5].

On the hardware side, these challenges are usually countered by building multiple cameras with different lenses into a single phone. While selecting a single, most suitable camera for each task solves some of the focal problems, algorithmic combination of multiple images form multiple time steps and cameras allows computational solutions for a wider range of problems:

◊ Low light image enhancement approaches like [14] already use neural networks to provide real-time high resolution images on mobile phones

◊ Burst denoising [28, 10] approaches use sequences of multiple images taken over time to compute a singe output image with increased overall exposure time. Apple's "Deep Fusion" [33] feature is a prominent example for this technique.

◊ Multi lens zoom [46] is another example where images from multiple cameras are algorithmically combined to allow better image magnifications and color consistency.

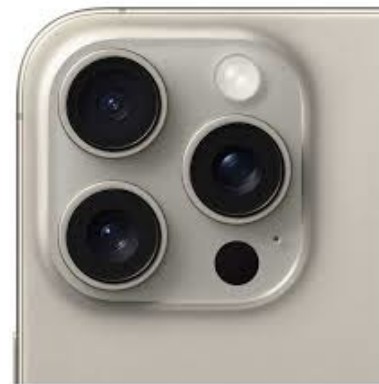

Figure 3: Camera Array of a recent *iPhone* 15Pro.

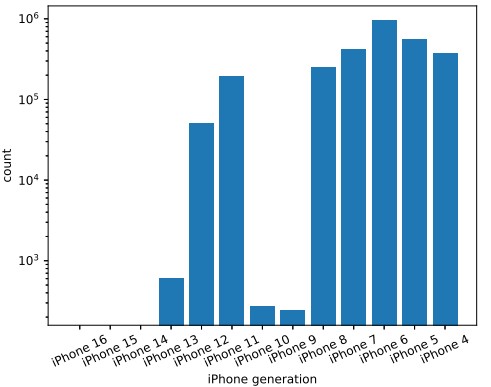
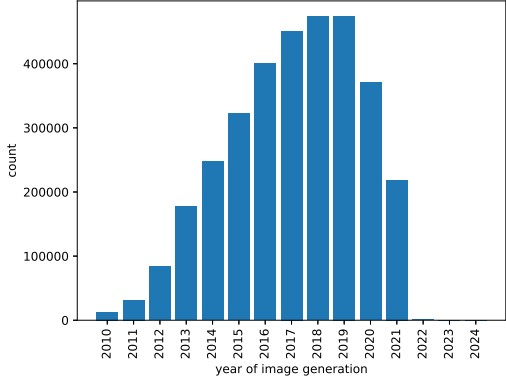

Figure 1: Novelty of photos and used imaging devices in the latest **Re-LAION 5B** dataset. For our *exif*-analysis we focus on images taken by *iPhones*, which make up about 20% of the images in the dataset. **Left:** distribution of *iPhones* models[1]. **Right:** distribution of the image generation dates. Note that even in this latest large scale image dataset, hardly any image is newer than 2021.

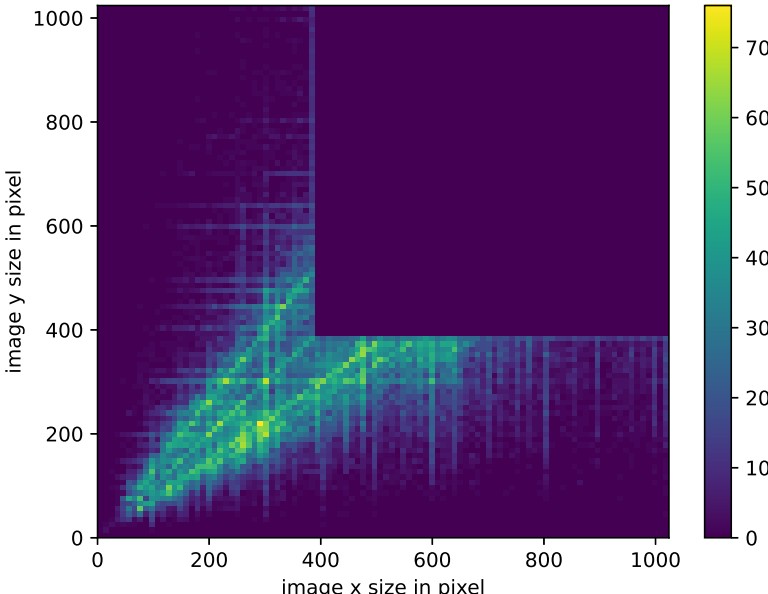

Figure 2: Size distribution in $x$ (width) and $y$ (height) of all LAION 5B images taken with *iPhones*. The cut of image resolutions at 400px of the smaller image dimension is a design choice of the dataset [38].

Overall, even so most phone manufactures provide very limited technical details beyond marketing claims, there are sufficient indications that many of these approaches are already in operation in modern consumer phones.

> ## Proposition #3
>
> **Modern imaging devices like mobile phones are computing rather than "taking" photos. "Real" training sets must contain such samples in order to generalize to real world scenarios.**
>
> - Per default, modern mobile phones use a wide range of image enhancement and fusion of multiples images taken from multiple cameras to compute photos.
> - The usage of these algorithms can only partially be controlled by the user, especially under poor lighting conditions.
> - Current fake-detection algorithms do not generalize to such images.

**Proof of concept I: Some detectors fail on images from modern phones.**

We back the key points of our proposition #3 with a series of experimental evaluations. First, we reproduce results of current SOTA detectors to establish a baseline. To this end we replicate the benchmark from [49], which provides a large collection of pre-trained detectors, datasets and a global evaluation script. The left plot in fig. 4 shows the aggregated results with the typical, well known generalization problems of some detectors [37, 32, 50]. The full detailed detector by dataset results are shown in tab. 3.

We extended these baseline experiments by two new evaluations: first we test the same models with 1000 random *iPhone* images taken from the Re-LAION-5B dataset, and second, with a few dozens RAW images manually taken with *iPhone* 13mini and 15Pro models (both using iOS 18.4.1).

The right plot in fig. 4 shows the combined results: the mostly down-sampled and compressed LAION-5B images appear to be well within the training distributions of most detectors (with an exception to DIRE) and are on par with the baseline or even outperform it. However, the RAW *iPhone* images cause a significant drop in accuracy for most detectors which indicates that modern imaging devices indeed produce out of distribution samples.

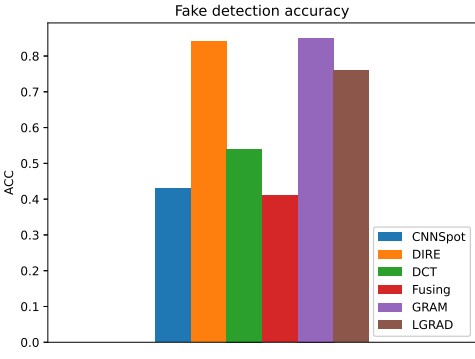 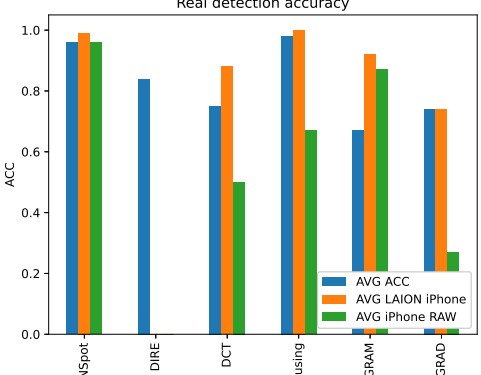

Figure 4: Overview of the results of the experimental evaluation of generated image detection algorithms. Details are shown in tab. 3. **Left:** Reproduction of the detection accuracy (ACC) of "fake" images using several detection algorithms the benchmark in [49]. Note: due to the long runtime, the results for DIRE are taken form [49]. **Right:** Evaluation of the detection ACC of the same algorithms for "real" images on three different benchmarks: **blue** reports the original benchmark used in [49], **orange** shows the same algorithms on *iPhone* images obtained from *LAION 5B*, **green** displays the detection ACC for raw images taken by *iPhone* 13 and 15pro models.

| DataSet | CNNSpot real | CNNSpot fake | DIRE real* | DIRE fake* | DCT real | DCT fake | Fusing real | Fusing fake | GRAM real | GRAM fake | LGRAD real | LGRAD fake |
|---|---|---|---|---|---|---|---|---|---|---|---|---|
| progan | 1.00 | 1.00 | 0.95 | 0.95 | 1.00 | 0.99 | 1.00 | 1.00 | 1.00 | 1.00 | 1.00 | 1.00 |
| stylegan | 1.00 | 0.74 | 0.83 | 0.83 | 0.88 | 0.68 | 1.00 | 0.70 | 1.00 | 0.67 | 0.98 | 0.82 |
| biggan | 0.94 | 0.47 | 0.70 | 0.70 | 0.69 | 0.95 | 0.94 | 0.61 | 0.46 | 0.89 | 0.78 | 0.86 |
| cyclegan | 0.92 | 0.79 | 0.74 | 0.74 | 0.77 | 0.80 | 0.95 | 0.79 | 0.63 | 0.85 | 0.89 | 0.83 |
| stargan | 0.97 | 0.86 | 0.95 | 0.95 | 0.90 | 1.00 | 1.00 | 0.94 | 1.00 | 1.00 | 1.00 | 0.96 |
| gaugan | 0.93 | 0.65 | 0.67 | 0.67 | 0.68 | 0.93 | 0.92 | 0.62 | 0.20 | 0.96 | 0.64 | 0.97 |
| stylegan2 | 1.00 | 0.69 | 0.75 | 0.75 | 0.92 | 0.40 | 1.00 | 0.67 | 1.00 | 0.72 | 0.99 | 0.73 |
| whichfaceisreal | 0.93 | 0.81 | 0.58 | 0.58 | 0.89 | 0.04 | 0.99 | 0.48 | 0.67 | 1.00 | 0.58 | 0.43 |
| ADM | 0.95 | 0.25 | 0.98 | 0.98 | 0.70 | 0.60 | 0.98 | 0.15 | 0.63 | 0.81 | 0.64 | 0.59 |
| Glide | 0.95 | 0.19 | 0.92 | 0.92 | 0.69 | 0.42 | 0.98 | 0.16 | 0.63 | 0.82 | 0.65 | 0.76 |
| Midjourney | 0.95 | 0.07 | 0.89 | 0.89 | 0.69 | 0.25 | 0.98 | 0.06 | 0.62 | 0.28 | 0.64 | 0.70 |
| stable_diffusion_v_1_4 | 0.95 | 0.07 | 0.91 | 0.91 | 0.69 | 0.11 | 0.98 | 0.04 | 0.63 | 0.95 | 0.63 | 0.63 |
| stable_diffusion_v_1_5 | 0.95 | 0.07 | 0.91 | 0.91 | 0.70 | 0.11 | 0.98 | 0.04 | 0.63 | 0.95 | 0.64 | 0.63 |
| VQDM | 0.95 | 0.15 | 0.91 | 0.91 | 0.69 | 0.89 | 0.98 | 0.12 | 0.63 | 0.80 | 0.64 | 0.73 |
| wukong | 0.95 | 0.07 | 0.90 | 0.90 | 0.69 | 0.14 | 0.98 | 0.06 | 0.62 | 0.87 | 0.63 | 0.53 |
| DALLE2 | 0.95 | 0.06 | 0.92 | 0.92 | 0.38 | 0.31 | 0.99 | 0.07 | 0.44 | 0.97 | 0.45 | 0.93 |
| **AVG ACC** | 0.96 | 0.43 | 0.84 | 0.84 | 0.75 | 0.54 | 0.98 | 0.41 | 0.67 | 0.85 | 0.74 | 0.76 |
| LAION 5B IPhone Images from 2010 | 0.99 | | 0.00 | | 0.88 | | 1.00 | | 0.84 | | 0.73 | |
| LAION 5B IPhone Images from 2021 | 0.99 | | 0.00 | | 0.89 | | 1.00 | | 0.97 | | 0.78 | |
| LAION 5B Images from iPhone4 | 0.99 | | 0.00 | | 0.85 | | 1.00 | | 0.91 | | 0.62 | |
| LAION 5B Images from iPhone12Pro | 0.99 | | 0.00 | | 0.89 | | 1.00 | | 0.98 | | 0.83 | |
| **AVG LAION iPhone** | 0.99 | | 0.00 | | 0.88 | | 1.00 | | 0.92 | | 0.74 | |
| IPhone 13mini good (iOS 18.4.1) | 0.89 | | 0.00 | | 1.00 | | 1.00 | | 1.00 | | 0.22 | |
| IPhone 13mini poor (iOS 18.4.1) | 1.00 | | 0.00 | | 0.50 | | 1.00 | | 1.00 | | 0.40 | |
| iPhone 15Pro (iOS 18.4.1) | 1.00 | | 0.00 | | 0.00 | | 0.00 | | 0.60 | | 0.20 | |
| **AVG iPhone RAW** | 0.96 | | 0.00 | | 0.50 | | 0.67 | | 0.87 | | 0.27 | |

Table 3: Detailed results of the experimental evaluation. Notes: *Due to the long runtime of a the DIRE approach we do not reproduce this experiment on the full dataset but report the original results from X which does only gives combined ACC values for fake and real.

**Proof of concept II: poor light conditions trigger enhancement and fusion, leading to dropping detection rates.**

For further analysis, we conduct a 4th experiment where we acquired RAW *iPhone 13mini* images of identical scenes under different illumination conditions to show the effect of automatic image fusion and enhancement algorithms. Fig. 5 shows an example scene and the 50% drop in accuracy for the evaluation of the DCT detector [13]. The full detailed detector by dataset results are shown in tab. 3.

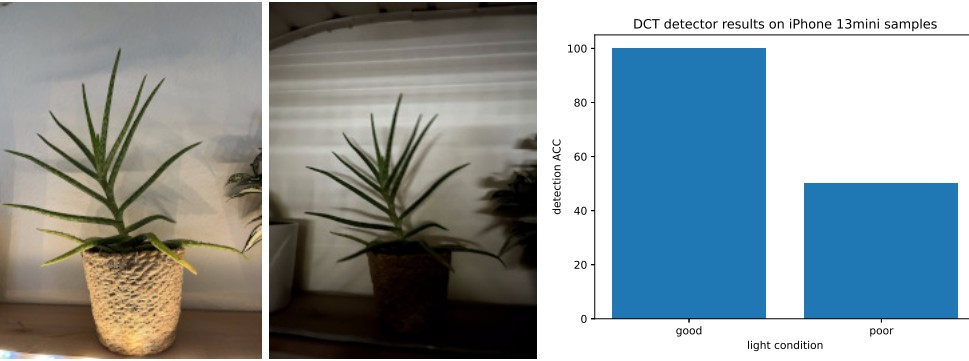

Figure 5: Effects of automatic image enhancements. **Left:** Example image taken with an *iPhone 13mini* (iOS 18.4.1) under good illumination conditions. **Center:** same scene taken by the same phone under poor illumination. **Right:** Impact of the the detection rate of the DCT detector [13], which utilizes low level image features in frequency space.

Of cause, our small proof of concept experiments can not provide a detailed and definite analysis of the described problems. For that purpose we would need a full-scale study on a large dataset containing modern phone images of various brands and models - which apparently does not (yet) exist. However, we argue that our small tests provide enough evidence to back our proposition.

> **Proposition #4**
>
> **The omnipresence of (automatic) image enhancing and fusion algorithms in modern imaging devices requires a redefinition of "real" images.**
>
> - All images are potentially subject to algorithmic processing which might even be in-transparent for the users.
> - With increasing computational power available in mobile imaging devices, these algorithms will further shift towards (generative) neural networks.
> - If essentially the same algorithms are applied to "real" and "fake" images, most feature and spectrum based low level detectors are prune to fail.
> - If all images are "manipulated", we need to redefine "fake" in a semantic rather than a technical manner.

Following the line of argumentation built in the previous propositions, we are facing the situation that the vast majority of images "taken" today, are processed by image enhancement and fusion algorithms. With increasing computational power [19] available in mobile imaging devices, these algorithms will further shift towards (generative) neural networks, while we can expect that their technical details and application will become more in-transparent for the user.

A direct consequence of this development is that we need to adapt the definition of "fake" images away from the simple technical *"has this image has been altered at all?"* towards a semantic *"has this image has been altered with a harmful intend?"*. Obviously, this change will make DeepFake detection a much harder problem. Most of the existing feature based approaches are not very likely to succeed in such a semantic settings and we need to discuss which alterations actually should be "allowed". The later is not going to turn out to be a trivial task, if it is possible at all: As the intentions

of image fakes are hard to infer from an image alone and the semantic impact of alteration heavily depends on the context, the same algorithmic image alteration might be uncritical in one image and producing a fake in another.

## Conclusions

Given our previous arguments, we derive the following conclusions:

### Conclusion #1

**We need new datasets for "real" images!**
The discussion of propositions #1-3 showed that current datasets are obviously not covering the distribution of images produced by modern imaging devises. If we further intend to engage the Deepfake problem by means of fake-detection algorithms, we at least need to be able to generalize them to the correct distribution.
While the collection of a new dataset appears to be manageable at first glance, we would like to point out several challenges that would need to be solved:

- The need for dynamic datasets which adopt to new imaging and image enhancement technologies over time.
- The difficult tradeoff between privacy and data protection on one side and the completeness of the of the "real" distribution on the other. For example, blurring all faces and license plates would be necessary to protect individual rights in a public dataset, but would fail to produce a correct estimate of the distribution of "real" images.

In our opinion, it would not be impossible, but take a large and long lasting effort to curate a sufficient "real" image dataset.

### Conclusion #2

**We need to agree which algorithmic enhancements are not altering reality...**
Following the argumentation in proposition #4, we need a new semantic definition of "real" images - otherwise all images are prune to become "fake" sooner or later. Hence, we need to somehow agree on which algorithmic enhancements are not altering the semantic meaning of images. Obviously such a definition will have to be context dependent and are thus very hard to agree upon. Unfortunately we are currently not able to even make a vague suggestion towards a solution. However, we strongly believe that the need for this discussion can not be ignored by the machine learning community.

### Conclusion #3

**We need to consider the possibility that DeepFake detectors are not a suitable solution.**
Following philosophical skepticism, we need to take the possibility into account that it actually might be impossible to give a consistent definition of "real" images (as proposed in Conclusion #2) and to correctly sample from its underlying distribution (as proposed in Conclusion #1). **If this would be the case, then any further research towards Deepfake detectors would be obsolete.**
This conclusion would not necessarily mean that no technical solutions are possible: there are already other approaches like watermarking and block-chain based image signatures which could allow implementations of alternative solutions of verifying "reality" and making image changes transparent.

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
