# OpenReview forum: "Deepfakes: we need to re-think the concept of “real” images"
_NeurIPS.cc/2025/Position_Paper_Track — Submitted to NeurIPS 2025 Position Paper Track_

### Official Review · Reviewer_Wz4H · 2025-07-23

**Significance:** 3
**Presentation:** 3
**Rating:** 5
**Confidence:** 3

**Summary:**

This position paper argues that current efforts in detecting fake/generated images neglect a critical issue: the lack of a clear and contemporary definition of “real” images. It highlights that most benchmark datasets used to train detection algorithms are outdated, low-resolution, and unrepresentative of modern imaging pipelines—particularly those produced by smartphone devices using image enhancement algorithms. Through a series of empirical evaluations and conceptual analysis, the authors call for a redefinition of what qualifies as “real” imagery, better datasets, and even question whether detection-based approaches are ultimately viable. The paper’s goal is to spark discussion and reflection in the NeurIPS community about foundational assumptions in deepfake detection research.

**Strengths:**

The paper challenges a widely held assumption in the deepfake detection community and raises a critical philosophical and practical issue. The four propositions are logically structured and well supported by experimental evidence and data analysis. The position is novel, timely, and of clear importance to both the machine learning and computer vision communities. The conclusion offers actionable insights, including the need for dynamic “real” image datasets and the reconsideration of semantic “realness” rather than technical authenticity.

**Weaknesses:**

The experimental evidence, while illustrative, is based on a limited sample (iPhone RAW images and LAION metadata) and may not be fully generalizable. The philosophical discussion could be more rigorous, and the paper does not engage deeply with other domains (e.g., legal, media theory) that have addressed notions of authenticity. Alternatives like watermarking are briefly mentioned but not analyzed in depth. Finally, there are some minor issues in phrasing and formatting throughout the paper.

**Questions:**

Could the authors elaborate on what criteria would be acceptable for defining "semantic authenticity" in images across domains (e.g., journalism, social media, surveillance)?

How do the authors envision building a community-curated, privacy-preserving dataset that adapts to evolving imaging technology?

Is there potential to learn "realness" as a latent concept from user-annotated data instead of defining it a priori?

**Alternative Position:**

Yes, and alternative positions are well-considered and addressed by the argument

**Author Identification:**

No.

**Context:**

3

**Discussion:**

3

**Ethics:**

["NO or VERY MINOR ethics concerns only"]

**Position:**

Yes, the paper argues for or against a position related to machine learning.

**Support:**

3

**Thoroughness:**

3

---

### Official Review · Reviewer_GgEM · 2025-08-08

**Significance:** 2
**Presentation:** 3
**Rating:** 4
**Confidence:** 3

**Summary:**

The position asserts that the current deepfake research domain is narrow in detection with the main recommendation being “rethinking” of strategy to include provenance tracking, watermarking, public awareness and multi-stakeholder collaboration.

**Strengths:**

S1: The position highlights a real-world, existing problem that detection lags behind generation.
S2: The authors frame the discussion as a societal harm, highlighting the importance of addressing this challenge.

**Weaknesses:**

W1: The discussion on legal and ethical trade-offs reads superficial, and a deeper discussion on this would be more useful for the position.
W2: The authors can include some new technical insights made through the position taken in this paper.
W3: The authors can include some statistics to provide evidence on the examples cited in the paper, such as detection accuracy trends.

**Questions:**

See weaknesses.

**Alternative Position:**

Yes, and alternative positions are well-considered and addressed by the argument

**Author Identification:**

No.

**Context:**

2

**Discussion:**

2

**Ethics:**

["NO or VERY MINOR ethics concerns only"]

**Position:**

Yes, the paper argues for or against a position related to machine learning.

**Support:**

2

**Thoroughness:**

4

---

### Official Review · Reviewer_sT64 · 2025-08-13

**Significance:** 3
**Presentation:** 3
**Rating:** 7
**Confidence:** 4

**Summary:**

The paper is a position piece arguing that progress in image deepfake detection is constrained not by the sophistication of generative models alone, but by an outdated and underspecified notion of what counts as a “real” image. The authors claim that most detection benchmarks rely on a small set of old, low-resolution “real” datasets (e.g., ImageNet, LSUN, CelebA), while the vast majority of photographs today are produced by smartphone pipelines that algorithmically compute imagery via multi-frame fusion and neural enhancement—processes that blur the boundary between “captured” and “generated.”

**Strengths:**

Well-articulated diagnosis of a real failure mode: detectors trained on stale “real” distributions may not generalize. The cross-check of Re-LAION-5B’s recency and resolution distribution (via EXIF and size histograms) is informative.

The survey table makes visible the limited diversity of “real” sources used by many benchmarks.

**Weaknesses:**

Empirical support is preliminary: the EXIF-based analysis is confined to iPhones as a proxy, and the proof-of-concept detector tests are small-scale (the paper itself notes the need for a “full-scale study”).

No concrete protocol for collecting, curating, and refreshing “real” datasets is offered beyond high-level desiderata (privacy, dynamism, coverage).

The semantic redefinition of “fake” (harmful intent) is compelling but under-operationalized—how would benchmarks annotate “harm,” and at what unit (pixel, object, narrative)?

**Questions:**

Please specify a minimal viable “Real-2025” schema: device diversity (brands, models, lens types), capture contexts (lighting, HDR, burst, night modes), resolution policy (retain native full-res), metadata policy (EXIF preservation), and refresh cadence (e.g., quarterly device updates). Also outline a privacy pathway (face/license plate handling) that balances legal obligations with distributional fidelity—this tension is acknowledged but not resolved.

Demonstrate which on-device steps most confound detectors (e.g., multi-frame HDR, denoising, upscaling, Deep Fusion-style burst fusion). A modular study (simulate each step) would inform detector design and could suggest augmentation strategies for robustness.

**Alternative Position:**

Yes, and alternative positions are well-considered and addressed by the argument

**Author Identification:**

No.

**Context:**

3

**Discussion:**

3

**Ethics:**

["NO or VERY MINOR ethics concerns only"]

**Position:**

Yes, the paper argues for or against a position related to machine learning.

**Support:**

3

**Thoroughness:**

4

---

### Note · Authors · 2025-09-02

**1-10 Additional Comments:**

We would prefer a standard rebuttal phase. Only being able to respond to the AC in very short 2k characters is a very limited possibility to rebuttal the reviews.

**1-11 Submit Again:**

Probably yes

**1-1 Submission Process:**

4

**1-2 Next Year:**

We think that position papers are a very suitable approach to engage the raising non-technical challenges in ML/AI research.  On one hand, we need citable problem statements to which authors can refer, on the other hand (and even more importantly), we as a research community need to engage (multidisciplinary) discussions about ethical and social impacts. Here it is key to find appropriate and generally accepted forums to publicly continue the discussion triggered by the position papers. -> see 1-4

**1-3 Future Development:**

It will probably take a few iterations to find an optimal setting for the new track (just like the dataset+benchmark track), but eventually this will probably converge to a generally accepted format. Please give this new format some time to develop, especially regarding the discussion formats -> see 1-4

**1-4 Interest:**

["Panel discussions with other position paper authors", "Workshops for developing position papers", "Other (please specify in the next question)"]

**1-4 Other Interest:**

The “birds-of-a-feather/” (BOF) format is well established in some communities -> see the HPC community as an example: https://sc25.supercomputing.org/program/birds-of-a-feather/

One idea could be to group several position papers into BOF sessions with short impulse presentations by  the authors, followed by a general discussion. A in-person BOF at the conference could be extended by an asynchronous discussion forum afterwards.

**1-5 Thoughtful:**

8

**1-6 Supportive:**

7

**1-7 Technical Aspects Versus Position:**

4

**1-8 Gate Keeping:**

10

**1-9 Camera Ready Changes:**

Based on the reviews and our own critical re-evaluation, we will focus changes to the camera ready version on the extension of the final discussion section. Both reviewer 1 and 3 have made the valuable suggestion that we should provide a clearer outline of proposed actions and add more detailed suggestions towards the solution of the presented problem. Hence, we will add:

* A brief outline to our proposed two step plan for the creation of new “real” datasets in the last section of the paper. First, we will suggest to create a smaller minimal viable “Real-2025” dataset for the systematic empirical analysis of the “real data” problem, and second, a larger “in the wild” dataset via crowd-sourcing.

* A detailed description of the minimal valuable “Real-2025” acquisition setup in the appendix.

We propose to collect the minimal “Real-2025” dataset from static scenes under controlled lab conditions in a miniature “model world” setup. Using a variety of different phones, software configurations and acquisition parameters, we would collect raw, high resolution images under varying light and motion conditions. The main challenge is, that most phones allow very little user control over the used enhancement algorithms (or their parameters). Hence, this has to be sampled implicitly, e.g. by a controlled manipulation of the acquisition environment (light, movement).

In a second step, we suggest to initiate a crowd-sourcing project for the collection of “‘real’ data in the wild”, following the privacy procedures proposed by the LAION project. However, a successful crowd data collection would require a broad and long lasting community effort which is only likely to succeed if the urgent need for such data is generally accepted. Hence, we would propose to postpone this step until this stage is reached (with help of this paper and the following discussions, further enhanced by the results of the proposed minimal viable dataset.

**3-1 Review Response1:**

sT64

**3-2 Reaction To Review1:**

We thank the reviewer for this detailed and thoughtful feedback. We agree with most points, as they exclusively concern possible (future) approaches to mitigate the problem of “real images”.

W1: Due to the lack of publicly available “real” images from mobile phones, our line of argumentation is deducted from an analysis and re-interpretation of existing literature, backed by a small empirical proof-of-concept. We agree that a first step towards a solution has to be a systematic, large scale empirical analysis. We will propose a more detailed setup for such a study in the final paper (see camera ready changes for details).

W2: We agree that the second step towards a solution would be an attempt to collect “real” data in the wild. Given the legal and technical problems known from previous web-crawling attempts, we would argue in favor of a crowd-sourcing approach  (see camera ready changes for details).

W3: We agree that a useful definition and annotation of “harmful fakes” is very hard. One possible approach could be a large scale interdisciplinary experiment which studies the human perception of image changes. Using latent space representations and application context, it might be possible to learn an implicit "harmfulness" measure.

Q1: As discussed in W1+W2, we propose to follow two approaches: first, a smaller minimal viable “Real-2025” dataset for the systematic empirical analysis of the “real data” problem, and second, a larger “in the wild” dataset (see camera ready changes for details).

Q2: Here we would propose to follow the protocol established for the re-LAION 5B dataset (https://laion.ai/blog/relaion-5b/) which provides a good blueprint for a privacy pathway.

Q3: Since we are not able to control the enhancement algorithms, it is practically impossible to conduct such a study directly. Instead, we suggest to apply such enhancement algorithms externally on unprocessed images and study their effects on the detection algorithms.

**3-3 Review Response2:**

GgEM

**3-4 Reaction To Review2:**

We thank the reviewer for the feedback. From reading the very brief summary and the three listed short weaknesses, we overall have the impression that the reviewer did not clearly understand the purpose of a position paper in general or the specific positions raised by our paper. The three named weaknesses are quite generic and only partially suitable for the evaluation of a position paper.

W1: The discussion of legal and ethical trade-offs of deep fakes is not the focus of our position. We don’t believe that more details toward these aspects would help to foster the discussion about our positions on the shortcoming of “real” data currently used in the context of deep fakes (and their detection).

W2: Unfortunately, it is not clear what new technical insights are missing in the reviewer’s opinion. Given that this is a position paper, we provide rather deep technical details about the shortcomings of current datasets and current image enhancement algorithms in phone imaging pipelines.

W3: Unfortunately it is not clear to which examples the reviewer is referring to.

**3-5 Review Response3:**

Wz4H

**3-6 Reaction To Review3:**

We would like to thank the reviewer for the thoughtful comments. We agree with many of the remarks, as they mostly focus on additional and followup works towards the solution of the proposed problem.

W1: We agree that a first step towards a solution has to be a systematic, large scale empirical analysis using many different hardware configurations (phones from different brands) and software stacks. We will propose a more detailed setup for such a study in the final paper (see camera ready changes for details).

W2: We agree that a much wider and multidisciplinary discussion is needed. Our aim is to initiate such a process. However, we think that the best way to convince the ML/AI community that such a process is needed, is to show the strong technical limitations of current technical solutions. We would argue that our position paper makes this point and thus has the potential to trigger such follow up-works from the ignited discussions.

W3: We agree that alternatives like camera based watermarking or block-chains should be part of a detailed discussion of possible solutions. However, in the context of the proposed positions, this is rather a side-aspect and we would argue that the limited space of the paper should be dominantly filled with the key arguments of the position.

Q1: We think that the implicit latent space based solution, which was also proposed by the reviewer in Q3) could be a promising approach. An explicit formulation of very application and context dependent criteria probably would be too complicated and static.

Q2: We propose a two step approach: first, the creation of a smaller dataset in a controlled lab environment, followed by a crowd sourcing of “real images in the wild”  (see camera ready changes for details).

Q3: Indeed, we agree that a large scale study on the human perception of certain image changes could lead the way towards an abstract latent space defined measure of "harmfulness" of changes to (or generations of) images.

---

### Meta-Review · Area_Chair_63nW · 2025-08-31

**Rating:** 5
**Confidence:** 4

**Strengths:**

This paper raises an interesting issue, what is real and fake in today's smartphone-dominated world, where most images pass through some AI system.
The paper goes through a careful empirical study demonstrating how the datasets used for deepfake detection do not reflect the current levels of AI processing of modern phone. This is most extreme in low-SNR imaging.
Deepfake detection is important to society.

**Weaknesses:**

While this paper does a good job of establishing the problem (which is not unknown), it does not sufficiently try to explain how to address it. Current benchmark are proposed on a best-effort basis. Such a position paper is incomplete without including possible solutions.
Reviewers though the philosophical, ethical and legal discussions needed to be more in-depth and rigorous. The AC is a nice-to-have, but not a complete blocker./

**Questions:**

Can you provide more detailed ideas for how to overcome the limitations of existing datasets? Perhaps the designation of fake or not is  not detailed enough, perhaps you can provide a more relevant taxonomy that will fit better with modern image pipelines?

**Thoroughness:**

3

---

### Decision · Program_Chairs · 2025-09-26

Reject